# Syncope Time Frames for Adverse Events after Emergency Department Presentation: An Individual Patient Data Meta-Analysis

**DOI:** 10.3390/medicina57111235

**Published:** 2021-11-12

**Authors:** Ludovico Furlan, Lucia Trombetta, Giovanni Casazza, Franca Dipaola, Raffaello Furlan, Chiara Marta, Filippo Numeroso, Jordi Pérez-Rodon, James V. Quinn, Matthew J. Reed, Robert S. Sheldon, Win-Kuang Shen, Benjamin C. Sun, Venkatesh Thiruganasambandamoorthy, Andrea Ungar, Giorgio Costantino, Monica Solbiati

**Affiliations:** 1Fondazione IRCCS Ca’ Granda Ospedale Maggiore Policlinico, Pronto Soccorso e Medicina d’Urgenza, 20122 Milan, Italy; furlan.ludovico@gmail.com (L.F.); chiara.marta@policlinico.mi.it (C.M.); giorgio.costantino@unimi.it (G.C.); 2Scuola di Specializzazione in Medicina Interna, Università degli Studi di Milano, 20122 Milan, Italy; lucia.trombetta@unimi.it; 3Dipartimento di Scienze Cliniche e di Comunità, Università degli Studi di Milano, 20122 Milan, Italy; giovanni.casazza@unimi.it; 4Department of Biomedical Sciences, Humanitas University, Internal Medicine, Humanitas Clinical and Research Center-IRCCS, 20089 Rozzano, Italy; franca.dipaola@humanitas.it (F.D.); raffaello.furlan@hunimed.eu (R.F.); 5Emergency Department, University Hospital of Parma, 43126 Parma, Italy; fnumeroso@gmail.com; 6Arrhythmia Unit, Department of Cardiology, Hospital Universitari Vall d’Hebrón, Universitat Autònoma de Barcelona, 08035 Barcelona, Spain; jordiperez@vhebron.net; 7Department of Emergency Medicine, Stanford University, Stanford, CA 94305, USA; quinnj@stanford.edu; 8Emergency Medicine Research Group Edinburgh (EMERGE), Department of Emergency Medicine, Royal Infirmary of Edinburgh, Edinburgh EH16 4SA, UK; Matthew.Reed@nhslothian.scot.nhs.uk; 9Libin Cardiovascular Institute of Alberta, University of Calgary, Calgary, AB T2N 4N1, Canada; Sheldon@ucalgary.ca; 10Mayo Clinic Arizona, Phoenix, AZ 85054, USA; wshen@mayo.edu; 11Department of Emergency Medicine, Leonard Davis Institute of Health Economics, University of Pennsylvania, Philadelphia, PA 19104, USA; Benjamin.Sun@pennmedicine.upenn.edu; 12Department of Emergency Medicine, University of Ottawa, Ottawa, ON K1Y4E9, Canada; vthirug@ohri.ca; 13Intensive Care Unit, Department of Geriatrics, University of Florence, Azienda Ospedaliero Universitaria Careggi, 50134 Firenze, Italy; andrea.ungar@unifi.it

**Keywords:** syncope, emergency department, outcomes, adverse events, arrhythmia, mortality, ECG monitoring, individual patient data, meta-analysis

## Abstract

*Background and Objectives*: Knowledge of the incidence and time frames of the adverse events of patients presenting syncope at the ED is essential for developing effective management strategies. The aim of the present study was to perform a meta-analysis of the incidence and time frames of adverse events of syncope patients. *Materials and Methods*: We combined individual patients’ data from prospective observational studies including adult patients who presented syncope at the ED. We assessed the pooled rate of adverse events at 24 h, 72 h, 7–10 days, 1 month and 1 year after ED evaluation. *Results*: We included nine studies that enrolled 12,269 patients. The mean age varied between 53 and 73 years, with 42% to 57% females. The pooled rate of adverse events was 5.1% (95% CI 3.4% to 7.7%) at 24 h, 7.0% (95% CI 4.9% to 9.9%) at 72 h, 8.4% (95% CI 6.2% to 11.3%) at 7–10 days, 10.3% (95% CI 7.8% to 13.3%) at 1 month and 21.3% (95% CI 15.8% to 28.0%) at 1 year. The pooled death rate was 0.2% (95% CI 0.1% to 0.5%) at 24 h, 0.3% (95% CI 0.1% to 0.7%) at 72 h, 0.5% (95% CI 0.3% to 0.9%) at 7–10 days, 1% (95% CI 0.6% to 1.7%) at 1 month and 5.9% (95% CI 4.5% to 7.7%) at 1 year. The most common adverse event was arrhythmia, for which its rate was 3.1% (95% CI 2.0% to 4.9%) at 24 h, 4.8% (95% CI 3.5% to 6.7%) at 72 h, 5.8% (95% CI 4.2% to 7.9%) at 7–10 days, 6.9% (95% CI 5.3% to 9.1%) at 1 month and 9.9% (95% CI 5.5% to 17) at 1 year. Ventricular arrhythmia was rare. *Conclusions*: The risk of death or life-threatening adverse event is rare in patients presenting syncope at the ED. The most common adverse events are brady and supraventricular arrhythmias, which occur during the first 3 days. Prolonged ECG monitoring in the ED in a short stay unit with ECG monitoring facilities may, therefore, be beneficial.

## 1. Introduction

Syncope is a frequent cause of emergency department (ED) visits and can be the final common presentation of numerous conditions, spanning from benign to life-threatening diseases [1]. Although numerous attempts have been made to develop consensus papers, risk prediction tools and clinical guidelines, the optimal management of patients with syncope in the ED is still uncertain [1,2,3,4,5].

Knowledge of the incidence and time frames of short-term outcomes after syncope is essential for informing effective management strategies and for supporting ED physicians in their decisions to discharge, admit or monitor patients [6]. Indeed, knowing the type and timing of short-term adverse events is essential in order to decide how long and in which setting we should observe patients [7]. Furthermore, it might aid decision making with respect to hospital admission and syncope units [8,9,10]. Moreover, assessing long-term outcomes after syncope might better inform differences between short and long-term risk factors and the possible role of different approaches (e.g., hospital admission and proper follow-up) for improving patients’ outcomes [10,11]. However, the relatively low incidence of adverse events and the heterogeneity in syncope definition, data collection and outcomes definition and assessment render it it difficult to assess the incidence and time interval between syncope and each adverse event [12,13,14].

The aim of the present study was to perform a meta-analysis of individual patients’ data in order to describe the incidence and time frames of short-term and long-term adverse events in patients presenting syncope at the ED.

## 2. Materials and Methods

### 2.1. Inclusion Criteria

We aimed to combine individual-level patient data from prospective observational studies by recruiting consecutive adult (i.e., ≥18 years old) patients presenting syncope at the ED. We asked international syncope experts (members of the scientific committee and invited participants of the First and Second International Workshops on Syncope First Assessment) if they were willing to share data from their studies, and if they were aware of other possible studies to be included.

In order to be included, studies had to report 7-day follow-up data. The syncope definition used in the original studies was accepted. Studies were excluded if they had the following factors: (1) only enrolled patients with a history of recurrent syncope; (2) only enrolled patients with a specific syncope etiology; and (3) only assessed treatment efficacy.

### 2.2. Outcomes and Definitions

According to a previous consensus, we defined any of the following as serious outcomes [12]: (1) all-cause death; (2) ventricular arrhythmia (i.e., ventricular fibrillation, sustained ventricular tachycardia and symptomatic non-sustained ventricular tachycardia); (3) supraventricular arrhythmias; (4) sinus arrest with cardiac pause > 3 s; (5) sick sinus syndrome with alternating bradycardia and tachycardia; (6) second-degree type 2 or third-degree AV block; (7) permanent pacemaker (PM) or implantable cardioverter defibrillator (ICD) malfunction with cardiac pauses; (8) ischemic or structural heart disease (defined as aortic stenosis with valve area ≤1 cm^2^, hypertrophic cardiomyopathy with outflow tract obstruction and left atrial myxoma or thrombus with outflow tract obstruction); (9) pulmonary embolism; (10) aortic dissection; (11) occult hemorrhage or anaemia requiring transfusion; (12) syncope resulting in major traumatic injury (trauma that requires admission or procedural/surgical intervention); (13) PM or ICD implantation; and (14) cardiopulmonary resuscitation. For the present study, we defined “arrhythmia” as any of the following: ventricular arrhythmia; supraventricular arrhythmias; sinus arrest with cardiac pause > 3 s; sick sinus syndrome with alternating bradycardia and tachycardia; and second-degree type 2 or third-degree AV block. We also defined as “bradyarrhythmia” any event with cardiac pause > 3 s; sick sinus syndrome with alternating bradycardia and tachycardia; and second-degree type 2 or third-degree AV block.

In cases where more than one outcome occurred in a patient, we considered only the first outcome, except for diagnoses and procedures where we included both outcomes. For example, if a patient underwent pacemaker implantation after a diagnosis of sick sinus syndrome, both adverse events were counted. We also collected the number of patients with at least one serious adverse event.

All the above outcomes were assessed 24 h, 72 h, 7–10 days, 1 month and 1 year after ED evaluation. We considered the cumulative incidence of outcomes (e.g., events happening at 7 days included both 24 h and 72 h events).

### 2.3. Statistical Analysis

For each study, we collected data on patient characteristics (mean age, sex and admission to hospital rate). The event rate with 95% confidence interval (CI) was calculated for every outcome at each pre-defined time frame. Due to expected clinical and methodological differences between the primary studies, all meta-analyses were performed by using the random effects model on the logit transformed rates. Pooled rates were calculated for each outcome at each time point. Statistical heterogeneity was quantified using the inconsistency index (I2) statistic, which ranges from 0% to 100% and is defined as the percentage of the observed between-trial variability that is due to heterogeneity rather than chance. All analyses were performed using SAS (release 9.4) statistical software.

## 3. Results

### 3.1. Study Selection and Characteristics

Among the members of the scientific committee and invited participants of the First and Second International Workshops on Syncope First Assessment, the authors of nine studies agreed to share individual patients’ data [11,15,16,17,18,19,20,21,22,23].

The included studies enrolled 12,269 patients, and the number of patients enrolled in each study varied between 312 and 6454. The mean age of patients varied between 53 and 73 years, and 52% of them were females (range 42% to 57%). Hospital admission rate was 32% with high heterogeneity between studies (range 12% to 83%). The studies were published between 2007 and 2020. Most were conducted in Europe (four in Italy, one in Spain and one in the United Kingdom), and two were conducted in the USA and one in Canada. Table 1 describes the main characteristics of the included studies.

### 3.2. Data Synthesis

The proportion of patients with adverse events varied between 2% and 13% at 24 h; 4% and 14% at 72 h; 4% and 16% at 7–10 days; 7% and 17% at 1 month; and 14% and 30% at 1 year. The pooled rates of patients with adverse events ranged between 5.1% (95% CI 3.4% to 7.7%; I2 94%; 11,653 patients; and 8 studies) at 24 h and 21.3% (95% CI 15.8% to 28.0%; I2 94%; 2917 patients; and 5 studies) at 1 year. Table 2 reports the pooled rates of patients with adverse events at the different time frames. Figure 1 shows the pooled rates of patients with adverse events over time.

Death was reported as a primary outcome in all studies. The pooled death rate was 0.2% (95% CI 0.1% to 0.5%) at 24 h; 0.3% (95% CI 0.1% to 0.7%) at 72 h; 0.5% (95% CI 0.3% to 0.9%) at 7–10 days; 1% (95% CI 0.6% to 1.7%) at 1 month; and 5.9% (95% CI 4.5% to 7.7%) at 1 year (Table 3 and Figure 2, panel A).

Arrhythmia was most common adverse event reported. The pooled rate of arrhythmia was 3.1% (95% CI 2.0% to 4.9%; I2 91%; 11,653 patients; and 8 studies) at 24 h; 4.8% (95% CI 3.5% to 6.7%; I2 88%; 10,553 patients; and 7 studies) at 72 h; 5.8% (95% CI 4.2% to 7.9%; I2 91%; 11,284 patients; and 8 studies) at 7–10 days; 6.9% (95% CI 5.3% to 9.1%; I2 90%; 11,175 patients; and 8 studies) at 30 days; and 9.9% (95% CI 5.5% to 17.1%; I2 96%; 2917 patients; and 5 studies) at 1 year (Table 3 and Figure 2, panel B).

The pooled rates of PM or ICD implant ranged from 0.6% (95% CI 0.3% to 1.4%; I2 78%; 10,593 patients; and 6 studies) at 24 h to 3.0% (95% CI 2.0% to 4.5%; I2 89%; 10,804 patients; and 7 studies) at 1 month and to 3.6% (95% CI 2.5% to 5.2%; I2 73%; 2917 patients; and 5 studies) at 1 year (Table 3 and Figure 2, panel C).

Data on the pooled estimates of the other outcomes at the different time frames are reported in Appendix A.

## 4. Discussion

In this study, we combined individual-level data from nine studies and 12,269 patients to describe the incidence and time frames of short-term and long-term adverse events in patients presenting syncope at the ED. This study confirms that syncope is potentially more benign than historically thought and that short-term mortality and life-threatening conditions (such as ventricular arrhythmias, pulmonary embolism and aortic dissection) are relatively rare. Adverse events mainly include arrhythmic (bradyarrhythmias and supraventricular arrhythmias), and these occur within the first 3 days after index syncope with incidence and does not increase much afterwards.

Knowledge of the outcomes of syncope patients may prove useful for improving the management of these patients [6]. Indeed, knowing the type and timing of short-term adverse events is essential for deciding how long and in which setting we should observe patients [7]. Knowledge of short and medium-term outcomes might aid decision making with respect to hospital admission and syncope units [8,9,10]. Previous attempts have been made to describe the incidence of adverse events after syncope by using meta-analyses [14,24]. However, the heterogeneity of both syncope and outcome definitions and the lack of standardized follow-up durations made it difficult to combine data from different studies. We tried to overcome this problem by combining individual-level patients’ data and by using homogeneous definitions for syncope and adverse events in order to obtain data on each single adverse event at short, medium and long-term follow-up.

The very low risk of short-term mortality, adverse outcome and high risk diagnosis supports data from previous studies [25], which have shown that these risks are low once patients with serious conditions already identified in the ED are excluded.

The proportion of patients with adverse events does not increase significantly between 72 h and 7–10 days after ED presentation. This suggests that most adverse events occur within the first 72 h and that extending observation beyond this time might not be worthwhile. A short period of observation in the ED or in a dedicated syncope observation unit could allow identification of most patients at risk of short-term adverse events. As most adverse events are arrhythmic, this study suggests that ECG monitoring during observation should be mandatory.

Previous studies have suggested that ECG monitoring should last 12 or 24 h and that this duration could allow identification of most of the patients with adverse events [7,21,26,27]. Our study, having analyzed not only the proportion of patients with arrhythmia but also the trend over time, suggests that 3-day observation with ECG monitoring will identify a significant proportion of patients with adverse events and that a longer observation period does not add much. After hospital discharge, ambulatory ECG monitoring and syncope units may help identify additional patients at risk of adverse outcomes.

As for the 1-year risk of death, our 5.9% rate is in line with previous systematic reviews showing a mortality rate of 8.4% [14] and 7% [24]. Unlike the other endpoints that show little increase after day 3, long-term mortality continues to rise. This finding might be due to several reasons. As mortality is a strong endpoint and does not depend on how thoroughly patients are investigated, we may be observing an underestimation of the incidence of other adverse events. Indeed, patients might have died of undiagnosed conditions during follow-up. Moreover, syncope might be the epiphenomenon of frailty and other comorbidities, and the increase in mortality is not due to syncope per se and is, thus, not influenced by ED management. Unfortunately, it is impossible to reliably investigate the causes of death. Moreover, we did not compare our mortality rates to those of the general population to assess if they are higher than the expected mortality.

Another limitation of the current study is that we adopted as adverse events what have previously been defined in international consensus documents [12,13]. However, some of them (such as arrhythmia) are dependent on how intensively patients are diagnosed; for others, the causal–effect relationship between syncope and the outcomes might be less certain [6].

Finally, although we were able to homogenize outcome definitions and time frames by analyzing individual-level patients’ data, we observed high heterogeneity among the included studies. This finding could be due to the inclusion of populations with different ages and admission rates and, therefore, contains different risk profiles. However, we cannot exclude the possibility that heterogeneity might be the result of different strategies in management and follow-up of patients. Indeed, as most of the events are non-life-threatening arrhythmias, a more aggressive diagnostic strategy could result in finding a higher number of outcomes. Nevertheless, it could unpredictably impact prognosis.

## 5. Conclusions

The findings of this study will help define the best approach for observing and ECG monitoring syncope patients in the ED, in short stay hospital facilities and as outpatients.

In patients presenting syncope at the ED, mortality and life-threatening adverse events are rare and high-risk diagnoses are usually recognized in the ED. Most of the adverse events at follow-up are brady and supraventricular arrhythmias and occur within the first 3 days. Therefore, if patients are admitted for observation, they should undergo ECG monitoring. Considering the time frame of serious outcomes, ECG monitoring should occur during the patient’s ED stay during a short admission to a unit with ECG monitoring facilities.

## Figures and Tables

**Figure 1 medicina-57-01235-f001:**
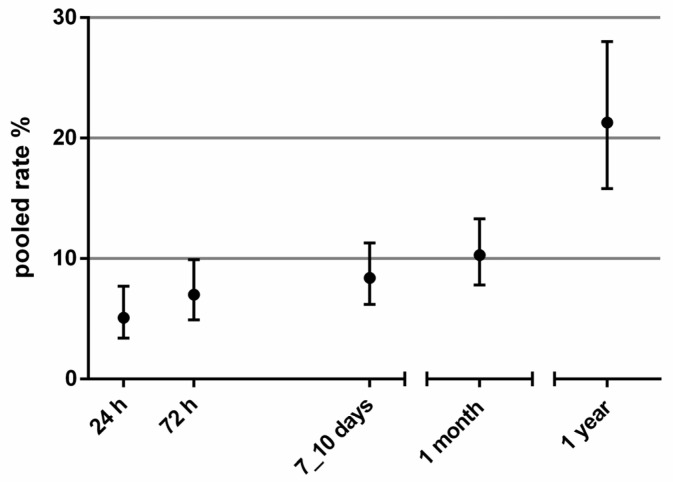
Pooled rates (with 95% CI) of patients with adverse events over time.

**Figure 2 medicina-57-01235-f002:**
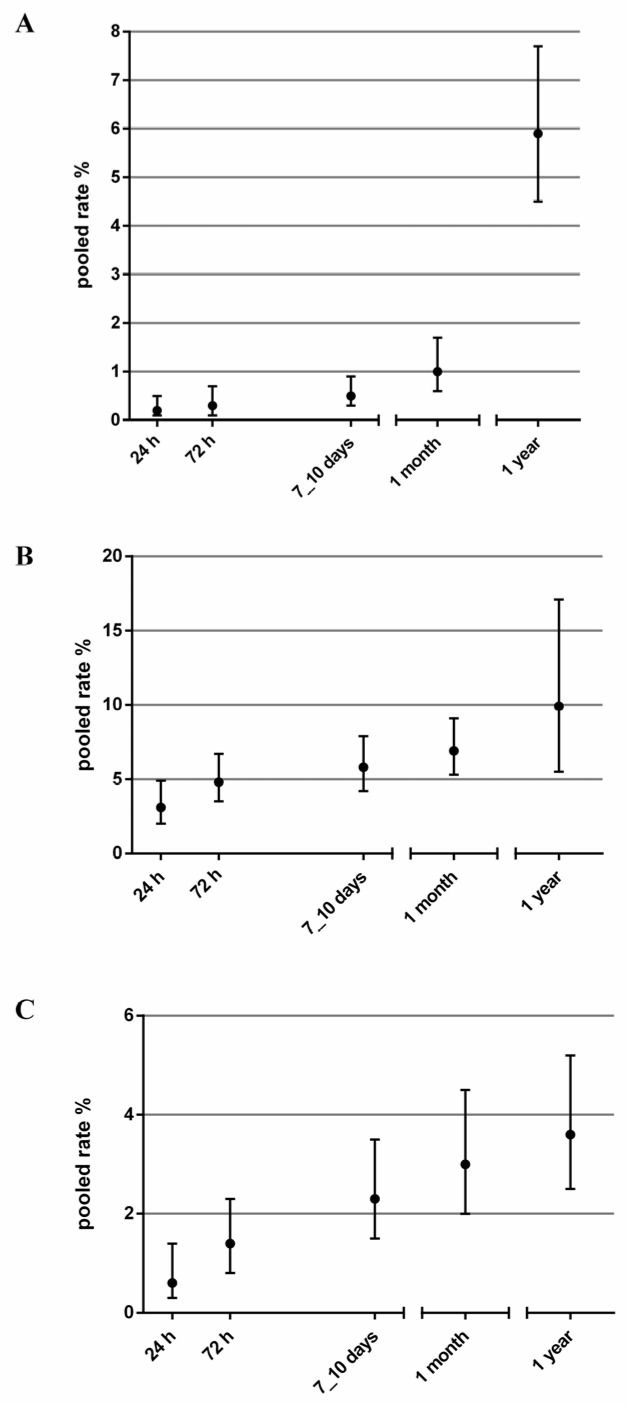
Pooled rates (with 95% CI) of mortality (**A**), arrhythmia (**B**) and PM or ICD implant (**C**) over time.

**Table 1 medicina-57-01235-t001:** Characteristics of the included studies.

Study	N of Patients	Mean Age (SD) (Years)	N of Females (%)	N of Admitted (%)
Sun_2007	312	56 (24)	179 (57)	182 (58)
Costantino_2008	689	64 (18)	386 (56)	236 (34)
Ungar_2010	371	66 (21)	168 (45)	144 (39)
Reed_2011	1100	63 (22)	587 (53)	538 (49)
Pérez-Rodon_2014	616	57 (20)	259 (42)	154 (25)
Numeroso_2016	347	73 (14)	162 (47)	157 (45)
Thiruganasambandamoorthy_2016	6454	53 (23)	3526 (55)	789 (12)
Solbiati_2020	345	65 (20)	171 (50)	102 (30)
Probst_2020	2035	73 (9)	989 (49)	1682 (83)

N: number; SD: standard deviation.

**Table 2 medicina-57-01235-t002:** Pooled rates of patients with adverse events at the different time frames.

	N of Studies	N of Patients with Events	N of Patients	Pooled Rate (%)	95% CI (%)	I2 (%)
24 h	8	545	11,653	5.1	3.4–7.7	94
72 h	7	602	10,553	7.0	4.9–9.9	93
7–10 days	8	766	11,284	8.4	6.2–11.3	94
1 month	8	902	11,175	10.3	7.8–13.3	93
1 year	5	599	2917	21.3	15.8–28.0	94

N: number; CI: confidence interval.

**Table 3 medicina-57-01235-t003:** Pooled rates of adverse outcomes at the different time frames.

		N of Studies	N of Events	N of Patients	Pooled Rate (%)	95% CI (%)	I2 (%)
Death							
	24 h	8	13	11,653	0.2	0.1–0.5	62
	72 h	7	19	10,553	0.3	0.1–0.7	72
	7–10 days	8	40	11,284	0.5	0.3–0.9	69
	1 month	8	81	11,175	1.0	0.6–1.7	80
	1 year	5	177	2917	5.9	4.5–7.7	66
Arrhythmia							
	24 h	8	388	11,653	3.1	2.0–4.9	91
	72 h	7	450	10,553	4.8	3.5–6.7	88
	7–10 days	8	582	11,284	5.8	4.2–7.9	91
	1 month	8	676	11,175	6.9	5.3–9.1	90
	1 year	5	282	2917	9.9	5.5–17.1	96
PM or ICD implant							
	24 h	6	86	10,593	0.6	0.3–1.4	78
	72 h	5	123	9493	1.4	0.8–2.3	79
	7–10 days	7	228	10,913	2.3	1.5–3.5	87
	1 month	7	273	10,804	3.0	2.0–4.5	89
	1 year	5	107	2917	3.6	2.5–5.2	73

N: number; CI: confidence interval; PM: pacemaker; ICD: implantable cardioverter defibrillator. Arrhythmia included ventricular arrhythmia (ventricular fibrillation, sustained ventricular tachycardia and symptomatic non-sustained ventricular tachycardia); supraventricular arrhythmias; sinus arrest with cardiac pause >3 s; sick sinus syndrome with alternating bradycardia and tachycardia; second-degree type 2 or third-degree AV block.

## Data Availability

Data sharing not applicable.

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
