# Peer review of "Syncope Time Frames for Adverse Events after Emergency Department Presentation: An Individual Patient Data Meta-Analysis"

_medicina, 2021, doi:10.3390/medicina57111235_

Round 1

Reviewer 1 Report

The paper is well written, easy to follow and addresses an interesting topic.  The introduction section could be more extended.

Reviewer 2 Report

The authors performed a meta-analysis concerning the incidence and time frames of adverse events in adult syncope patients who presented to the ED. The manuscript is well written and leads to reasonable conclusion. This information seems to be useful in the clinical practice, especially in the ED. However, there are a few concerns for acceptance.

  1. This study included adult patients who referred to the ED for syncope. Therefore, the authors need to present the study population in the title: e.g. an individual patient data in the ED meta-analysis.
  2. The definition of “arrhythmia” or “arrhythmic events” is unclear. It may be included “bradyarrhythmias and supraventricular arrhythmias” not included “ventricular tachyarrhythmias” as described page 7, line 166-167. Please make sure the definition of “arrhythmia” or “arrhythmic events” and describe it in the definition section of the manuscript.
